# Reflection in Rural Family Medicine Education

**DOI:** 10.3390/ijerph19095137

**Published:** 2022-04-23

**Authors:** Ryuichi Ohta, Chiaki Sano

**Affiliations:** 1Community Care, Unnan City Hospital, 699-1221 96-1 Iida, Daito-cho, Unnan 699-1221, Japan; 2Department of Community Medicine Management, Faculty of Medicine, Shimane University, 89-1 Enya cho, Izumo 693-8501, Japan; sanochi@med.shimane-u.ac.jp

**Keywords:** reflection, rural family medicine education, medical resident, medical teacher, nurses, community hospital, Japan

## Abstract

Reflection in medical education is vital for students’ development as professionals. The lack of medical educators in rural family medicine can impinge on the effective reflection of residents’ learning. Hence, based on qualitative research, we proposed a framework regarding reflection in rural family medicine education, indicating when, where, and how reflection is performed and progresses. The contents of reflection include clinical issues regarding knowledge and skills, professionalism in clinical decisions, and work-life balance. The settings of reflection include conference rooms, clinical wards, residents’ desks, and hospital hallways. The timing of educational reflection includes during and after patient examination and discussion with various professionals, before finishing work, and during “doorknob” times (right before going back home). Rural medical teachers need competence as clinicians and medical educators to promote learning in medical residents and sustain rural medical care. Furthermore, medical teachers must communicate and collaborate with medical residents and nurses for educational reflection to take place in rural family medicine education, especially regarding professionalism. In rural family medicine education, reflection can be performed in various clinical situations through collaboration with learners and various medical professionals, aiding the enrichment of residents’ learning and sustainability of rural medical care.

## 1. Introduction

Reflection is a vital part of family medicine education to facilitate trainees’ development as professionals [1]. Medical residents gather experience in various clinical situations to enhance their learning [2] by acquiring medical knowledge, skills, and attitudes. Educational reflection on these clinical learnings can motivate the trainees in various other situations [3]. In educational reflection, medical educators review residents’ experiences and communicate to them the different perspectives of residents’ learning, mistakes, and perceptions, and guide them toward the next steps of clinical practice [4]. Since family medicine education involves various experiences in different clinical settings, reflection should be made an integral part of these experiences [1,5].

In rural family medicine education, the lack of medical educators can adversely affect effective reflection [6,7]. As rural community hospitals lack medical professionals, rural family medicine educators also work as practitioners who treat patients in rural communities [8,9] and require them to effectively manage their time between clinical practice and medical education [10,11]. An extremely busy schedule may make it difficult for them to use their time with their residents appropriately [12,13], negatively affecting residents’ educational reflection. This makes it necessary to identify effective methods of reflection, especially in rural family medicine education.

In Japan, family medicine education had its official start in 2010. Hence, few family physicians practice in rural settings. In the Japanese healthcare system, patients can choose to visit any medical institution at any time, thus community hospitals have to effectively deal with the various symptoms they present. Consequently, rural healthcare has a high demand for family physicians to effectively approach various patient needs. Hence, rural family medicine education is essential for the sustainability of community care, with family physicians playing an important role in improving sustainability [14]. Rural medicine lacks medical professionals. Therefore, rural family medicine has the potential to improve patient care, making educational reflection increasingly crucial [15,16]. An effective reflection system can be essential to drive family medicine education. Currently, there is a lack of evidence regarding the method of reflection in rural family medicine. Therefore, this study aimed to investigate the framework of effective reflection in rural family medicine education.

## 2. Materials and Methods

This qualitative research was conducted to clarify if reflection in rural family medicine education improves residents’ development. Ethnography and interviews were conducted from 1 April 2020 to 31 December 2021.

### 2.1. Setting

The study setting was Unnan City Hospital, located southeast of Shimane prefecture, a rural Japanese prefecture. At the time of the study, the hospital had 281 care beds: 160 acute care, 43 comprehensive care, 30 rehabilitation, and 48 chronic care beds. The nurse-to-patient ratios were 1:10 for acute care, 1:13 for comprehensive care, 1:15 for rehabilitation, and 1:25 for chronic care. The hospital had 27 physicians, 197 nurses, 7 pharmacists, 15 clinical technicians, 37 therapists, 4 nutritionists, and 34 clerks.

The hospital had a rural family medicine education curriculum, which accommodated three family medicine educators. In this curriculum, residents experienced various clinical situations with their patients. In their first year, residents worked at the Unnan City hospital and treated typical diseases, in both inpatient and outpatient situations. In the following year, they worked at a rural clinic (Kakeya Clinic) for six months to learn home care and community-oriented primary care. To broaden their scope of practice in internal medicine, pediatrics, and emergency medicine, they worked at a general or community hospital for one and a half years.

Each clinical setting included a medical teacher. This curriculum can be utilized to educate up to three residents simultaneously. One resident in the years 2018 and 2019, and three in the years 2020 and 2021 engaged in the curriculum.

### 2.2. The Framework of Reflection

Effective reflection requires educators’ competence and collaboration with medical residents and nurses. In our research, educational reflection was performed according to Kolb’s Learning Cycle, based on the social cognitive theory [17]. Through clinical experiences, medical residents can realize their assets and drawbacks as physicians. Educational reflection can then enable concrete revision of their learning [18]. Further, the repetition of reflection facilitates medical residents to connect their experiences and learn better with abstract conceptualization (Figure 1).

### 2.3. Ethnography and Semi-Structured Interviews

The first author performed ethnography and semi-structured interviews with the participants. This researcher’s specialties are family medicine, medical education, and public health. The researcher worked in all hospital wards, observed the interactions between residents and nurses in each ward, and took field notes during this process. During the observation period, the researcher interviewed the participants at the end of each month. The interview guide included five questions: “How did you consider about the place of reflection?”, “How did you consider about the timing of reflection?”, “How did you consider about the contents of reflection?”, “How do you do reflection effectively in rural family medicine education?”, and “Do you have any idea to improve the quality of reflection in family medicine?” Each interview lasted about 30 min and was recorded and transcribed verbatim. the interviews were conducted in Japanese. The transcript was shared with the respective interviewee to confirm the credibility of its content.

### 2.4. Analysis

Thematic analysis was used. The first author carefully and thoughtfully read the field notes and interview transcriptions. After reading them in-depth, the first author coded the contents and developed codebooks based on the repeated reading of the research materials as the initial coding. This study used process and concept coding. The first author, thus, induced, merged, deleted, or refined the concepts and themes by going back and forth between the research materials and initial coding. Finally, the theory was discussed by both authors, who ultimately reached an agreement on the final themes. The themes and quotes were translated in English. Our research was approved by the Unnan City Hospital ethical committee (approval no. 20190005).

## 3. Results

Based on the thematic analysis, the framework of reflection in rural family medicine education comprised five themes: contents, place, timing, competence as clinicians and medical educators, and collaboration with medical residents and nurses. Effective educational reflection calls for the integration of the five themes (Figure 2). The following themes emerged in the framework based on the interview transcripts and existing research.

### 3.1. Contents

The process of educational reflection includes clinical issues regarding knowledge and skills, professionalism in clinical decisions, and work-life balance, to enhance residents’ development as physicians.

Medical residents face various difficulties in clinical situations and learn about medical knowledge and skills. While they can devise their own methods to learn, overwhelming situations may impinge on their learning processes [19]. One of the residents stated, “My learning method from the previous hospital could not be used for practice in this hospital. That may be because I had to learn about various diseases in different medical categories at the same time. The learning can be different from other specialties.” Medical educators help medical residents reflect on their clinical learning based on Kolb’s Learning Cycle [17], which help to ensure effective learning.

Professionalism is taught through the educational reflection of clinical decisions on various patient conditions. In rural contexts, there are several cases of older patients with multi-morbidities. Rural physicians often have to treat old patients with multi-morbidities and construct treatment plans considering not only patients’ medical issues but also their psycho-social and familial aspects [20]. One resident stated, “Clinical decisions [are] complicated in the care of older patients. I needed to enhance my professionalism through the reflection with medical teachers.” For effective communication of clinical decisions, medical residents need varied knowledge and skills to communicate with different healthcare professionals. Such clinical communication places an immense burden on the cognitive functions of medical residents [20]. One medical teacher stated, “We should reflect on the residents’ experiences related to professionalism because they may be confused in the complicated situations needing ethical decisions.” To promote learning and relieve residents’ mental stress, in interprofessional collaboration and communication with patients, family, and multiple professionals, medical educators should have various dialogues as a form of reflection. 

Work-life balance is an essential topic used by medical residents during reflection for both career and personal development as it facilitates flexible career paths and working effectively [21]. Rural family medicine residents are affected by various clinical and personal experiences, such as marriage and childbirth, which impact their future [14]. One resident stated, “I had a son and planned to [take] paternity leave. I had modified the training schedule. I was anxious about my [future].” They may, consequently, require modifications in family medicine education training. Through the reflection, medical teachers use the topic of work-life balance to mitigate residents’ anxieties. One medical teacher stated, “Modification of [the] training schedule is stressful for the residents. The reflection should deal with issues related [to] work-life balance.” Medical educators’ supportive attitudes mitigate residents’ anxieties, especially through educational reflection on the perceptions about their future.

### 3.2. Place

The settings where reflection took place included conference rooms, clinical wards, residents’ desks, and hospital hallways. As rural medical educators are often busy, they need to use various places for educational reflections. Usually, educational reflection should be performed in conference rooms and residents’ desks to separate their cognitive load from clinical situations [4]. Since rural medical educators also work as clinicians and follow the medical decisions of residents, clinical places such as wards and hospital hallways can be used for educational reflection. One medical teacher stated, “I have to use [my] time effectively with our residents. Any place can be used.”

Conference rooms and residents’ desks can be the most effective places for educational reflection, while also ensuring the medical residents’ work-life balance. In clinical situations, medical residents may prioritize patients’ concerns over their own long-term career goals or private lives. Additionally, their privacy should be protected from other colleagues and workers in medical institutions [22]. Private spaces should be used for educational reflection on private issues regarding work-life balance.

For rural family medicine educators, educational reflection in clinical situations can be useful to save time and impart education effectively. They hope to use their time effectively in clinical settings. Clinical issues can be dealt with in clinical situations by collaborating with medical residents [23]. Medical educators, along with medical residents, can manage patient conditions as well as conduct educational reflection in clinical settings. One resident stated, “Reflection in clinical wards is useful because I could use the contents of reflection soon.”

### 3.3. Timing

Reflection occurs during and after patient examinations and discussions with various professionals, before finishing work, and during “doorknob times” (i.e., right before going back home). Each of these can be used effectively for educational reflection.

Clinical issues should be reflected upon during or soon after clinical experiences for effective learning. Learning materials should be reviewed as soon as possible for this purpose [24]. Medical residents gain clinical experiences in various settings such as outpatients’ departure, emergency rooms, and inpatient situations. Medical educators visit these settings and communicate with medical residents about their experiences based on Kolb’s learning cycle. One medical teacher stated, “Clinical issues should be discussed with residents in clinical situations as soon as possible. Medical teachers can use time with residents by moving to their clinical situations.” Continuous experiences with medical educators enable residents to have transient reflections during and about their clinical experiences. These reflections can help them revise their medical practice [25].

Based on medical residents’ work-life balance, specific times, such as before finishing work and “doorknob” times, should be used for educational reflection regarding stress management and controlling cognitive load. Medical residents may be tired and may reflect on their learning only briefly soon after finishing work. In such situations, medical teachers can go to their desks and communicate with them, supporting their reflections [26] and asking residents about the difficulties faced during practice. In clinical settings, doorknob questions can be used when patients are leaving the examination room while holding the doorknob to investigate their true anxiety by asking questions at the end of the session [27]. Medical reflections could be performed at this time to inquire about residents’ anxiety in their clinical situations, as well as those regarding their work-life balance. As one of the residents stated, “Medical teachers’ support before finishing work is useful, and frequent communication with medical teachers can facilitate me to confess our difficulties to them easily”.

### 3.4. Competence as Clinicians and Medical Educators

Rural family medicine educators attempt to be competent, both as clinicians and medical educators. First, medical educators are thought to be good physicians as they are role models for medical residents. Medical residents tried to develop their competencies by imitating medical teachers’ behaviors as clinicians. One of the medical teachers stated, “Rural medical teachers should be competent to be medical doctors and teachers. In clinical settings, we have to display professional attitudes as physicians. We have many opportunities to interact with medical residents. We should use these opportunities effectively.” In rural contexts, with fewer healthcare professionals, the interaction between individual medical teachers and residents can be more frequent than in an urban context [14]. Therefore, the teacher’s competency as a clinician can be vital for residents’ education and educational reflection.

Furthermore, rural medical educators try to be open-minded and flexible in clinical and educational settings. Since rural medical teachers need to also work as clinicians, they must use their time efficiently. Moreover, they must educate their medical residents on the importance of educational reflection at various timing and places. A rural medical teacher stated, “We have to work in a clinical setting as well. So, as medical teachers, we should use various situations for education. By increasing the time with residents for reflection, we can learn a lot.” Rural medical teachers should adjust their schedules for medical residents and manage timing and places for educational reflection.

### 3.5. Collaboration with Medical Residents and Nurses

Medical educators communicated and collaborated with medical residents and other professionals to build good relationships, as well as for effective educational reflection. Continued reflection could improve their relationships, which could enhance the quality of their reflection. One resident stated, “I could communicate with medical teachers a lot. I [can] say various things frankly to them.” Continuous educational reflection is essential for rural family medicine education. For efficiency, medical teachers and residents adapt to each other’s schedules based on the good rapport they build.

Furthermore, the relationship between medical educators and nurses was established for effective rural family medicine education. Nurses are good educators for teaching professionalism and interprofessional collaboration Furthermore, they can facilitate change in medical residents [28]. Medical residents could learn these qualities through discussions with nurses in clinical situations. One resident stated, “I could discuss with nurses in the ward regarding patients’ care. They have different perspectives and taught me a way of effective dialogue with patients and families.” Medical residents must also modify their working styles to suit current work requirements. Rural nurses could teach rural hospital culture and mitigate medical residents’ work difficulties. One medical resident stated, “We had various difficulties regarding working at the rural hospital initially. Medical teachers’ reflection is beneficial. Dialogue with nurses as reflection can also be beneficial.” These roles could also facilitate educational reflections among medical educators and residents regarding professionalism and interprofessional collaboration. One medical teacher stated, “Nurses’ and residents’ communication can be helpful for me. Nurses’ feedback on residents based on the reflection could modify residents’ behaviors without medical teachers’ feedback.” The collaboration among medical teachers and nurses mitigated medical teachers’ time constraints in reflection.

## 4. Discussion

This qualitative research shows the reflection framework in rural family medicine education. For effective reflection, the contents, settings, and timing are vital. These components are intertwined. Medical educators must manage and develop their competencies as clinicians and educators. Further, through rural family medicine education, medical teachers could improve their clinical knowledge, skills, and attitudes as clinicians.

The enhancement of educational reflection in rural family medicine education has the potential to improve the quality of care in rural contexts. First, enhancement of educational reflection could improve the curriculum of rural family medicine education, which could be appealing to medical residents who want to become family physicians [15]. Second, an increase in the number of family medicine residents in rural community hospitals could enable sustainable medical care in rural contexts, which often lack medical resources [14]. Third, in the process of enhancement of educational reflection, various medical professionals, including physicians, medical residents, and nurses could communicate with each other, strengthening their collaboration [29]. Since better interprofessional collaboration improves patient care, this strengthened collaboration could improve rural patient care. As the learning from nurses and citizens can drive medical residents’ learning, they can be involved in the reflection process [28,30].

Rural medical educators could improve their abilities as both educators and clinicians. Through educational reflection, medical educators must manage various medical issues along with their residents by learning together as playing managers. Thus, they should be flexible enough to approach ethical and professional issues that medical residents face with other medical professionals [14,28]. Through this process, medical residents can improve not only medical knowledge and skills but also professional judgments and interprofessional collaboration. In rural settings, medical teachers as playing managers can be role models for family medicine education, which can improve residents’ motivation and clinical performance [31,32]. Currently, as there is a lack of rural medical educators, more rural clinicians should be motivated to become medical educators through government educational systems to sustain rural medical conditions.

This study has some limitations, the first of which is the relationship between researchers and participants. The researcher had an educator–learner relationship with the residents. Therefore, the participants may have felt uncomfortable raising concerns. To overcome this limitation, the researcher tried to perform the interviews after the evaluation of their practices for participants to not be conscious of their assessment. Another limitation is the locality. This research was performed in one rural hospital. Future studies should investigate effective reflection methods in other regions and international contexts. Additionally, the interview transcripts were coded by the first author, which could affect this study’s credibility. To improve the quality of research, the second author reviewed the process of coding, concepts, and themes as theoretical triangulation.

## 5. Conclusions

This study presents a framework for reflection in rural family medicine. In the framework, the contents, timing and place should be accounted for medical educators to effectively perform educational reflection. Medical educators’ ability regarding clinical practice, medical education, and collaboration with residents and nurses can be essential for sustaining reflection.

## Figures and Tables

**Figure 1 ijerph-19-05137-f001:**
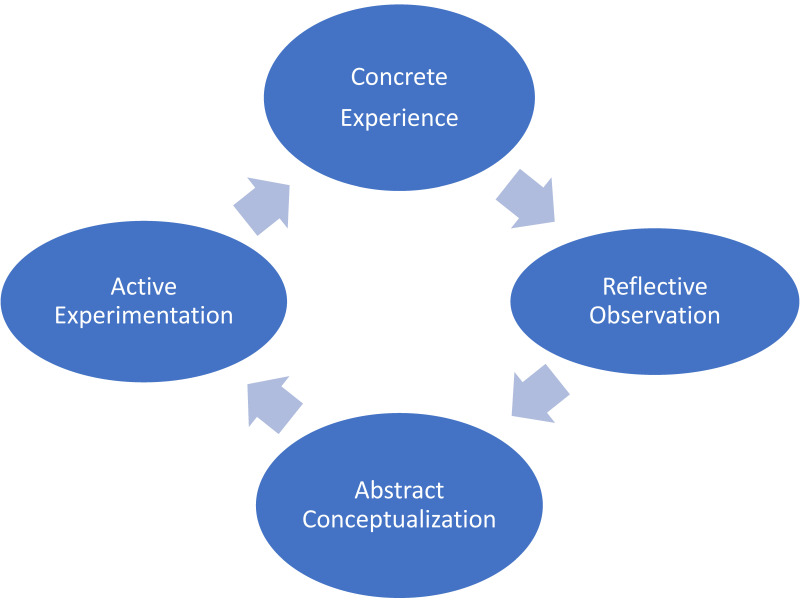
The framework of Kolb’s learning cycle.

**Figure 2 ijerph-19-05137-f002:**
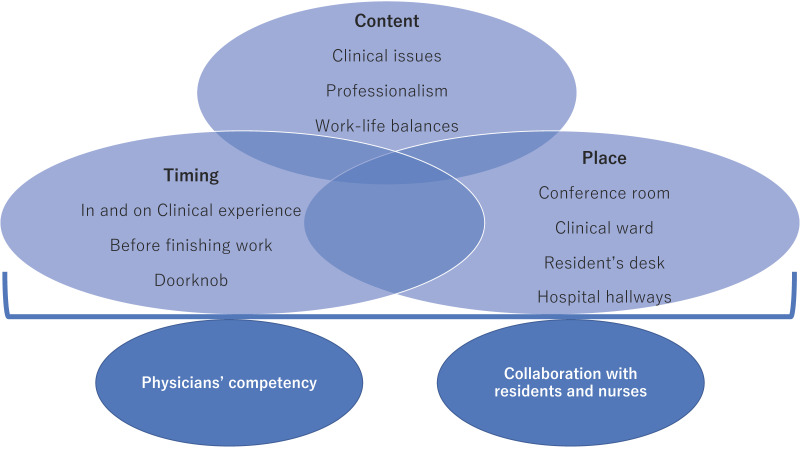
The framework of reflection in rural family medicine education.

## Data Availability

The datasets used and/or analyzed during the current study may be obtained from the corresponding author upon reasonable request.

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
