# Peer review of "Reflection in Rural Family Medicine Education"

_ijerph, 2022, doi:10.3390/ijerph19095137_

Round 1

Reviewer 1 Report

Dear author(s)

Thank you for your effort to discuss reflections in rural medical education. Overall, the manuscript is well-written and the framework you are proposing, can help medical educators improve the experience of residents and overall professionalism. However, I find the manuscript a little too theoretical with very few details given on how some of these components can be accomplished in real-life situations. Please use my comments, in the attached pdf file,  to add some of these details. 

Author Response

Response to Reviewers

Reviewer 1

Thank you for your effort to discuss reflections in rural medical education. Overall, the manuscript is well-written and the framework you are proposing, can help medical educators improve the experience of residents and overall professionalism. However, I find the manuscript a little too theoretical with very few details given on how some of these components can be accomplished in real-life situations. Please use my comments, in the attached pdf file, to add some of these details. 

Line 45

I am not sure what you are referring to by "the same"

Response: Thank you for the valuable feedback. We revised it to “sustainability.”

Line 75

This term is confusing, multimorbidity can be used if you mean patients with more than 2 conditions occurring at the same time. No need for "co-"

Same here, be consistent with comorbidities or multimorbidities

Response: We revised it to “multi-morbidities.”

Line 84 to 91

This paragraph is confusing, how does that relate to the content of the reflection? Please explain how will this be incorporated in the reflection. perhaps give examples of questions they may ask themselves.

Response: We now describe how work-life balance is related to the contents of reflection (Line 140-151).

Line 137

How is that measured or achieved in the proposed model?

Response:

We have revised our Methods section based on the research process with the pertinent quotations from participants (Line 53-103).

Line 151

This section is very theoretical. How can these collaborations take place, and what would be the role of the reflection to facilitate the collaborations?

Response:

Per your comment, we have revised our Results section based on the research process with pertinent quotations from participants (Line 104-247).

Line 162 to 167

how about time constraints, how can these be overcome?

Response: We revised our Results section to link the findings with our study process. We also included pertinent quotations from participants, including those discussing teachers’ time constraints.

Reviewer 2 Report

Thank you for the opportunity to review this paper. I found the topic interesting and believe it may be of interest to other readers of this journal. However, revisions are required before I feel this paper can be considered for publication.

The major issue I have with this paper is that there was no description of how this framework was developed. Section 2 (technical report) provides a brief summary of the three elements in the framework and mentions Kolb’s learning cycle and social cognitive theory. However, there are no details of the methods used to develop the framework. For instance, was a literature review conducted to identify relevant aspects of the framework? Were concept elicitation interviews conducted with relevant people to establish the elements? Or perhaps a Delphi panel? How was the validity of the framework assessed? Without this critical information, this work does not reflect a scientific study, rather the ideas of the authors. I do like the idea of the framework and feel such a framework could have value. However, if such a framework is to be useful it must be developed using sound scientific procedures, which are not evident in this paper. If a scientific method such as any of those described above were used, then they could be described in the paper and the paper could be reconsidered for review. If not, I encourage the authors to take a fresh approach to their enquiry using scientific principles.

A more minor point at this stage is that it was not clear if this framework is intended to only apply to one healthcare system (such as Japan, where the authors are based), or more globally. Details of the healthcare system in which this framework could be applied would be helpful for the readers of an international journal.

Author Response

Reviewer 2

Thank you for the opportunity to review this paper. I found the topic interesting and believe it may be of interest to other readers of this journal. However, revisions are required before I feel this paper can be considered for publication.

The major issue I have with this paper is that there was no description of how this framework was developed. Section 2 (technical report) provides a brief summary of the three elements in the framework and mentions Kolb’s learning cycle and social cognitive theory. However, there are no details of the methods used to develop the framework. For instance, was a literature review conducted to identify relevant aspects of the framework? Were concept elicitation interviews conducted with relevant people to establish the elements? Or perhaps a Delphi panel? How was the validity of the framework assessed? Without this critical information, this work does not reflect a scientific study, rather the ideas of the authors. I do like the idea of the framework and feel such a framework could have value. However, if such a framework is to be useful it must be developed using sound scientific procedures, which are not evident in this paper. If a scientific method such as any of those described above were used, then they could be described in the paper and the paper could be reconsidered for review. If not, I encourage the authors to take a fresh approach to their enquiry using scientific principles.

Response:

Thank you for the valuable feedback. We thoroughly revised the whole manuscript and strengthened our discussion of the research methods. We performed ethnography and interviews in this research, which we now describe in more detail in Section 2.3. Furthermore, we revised our Discussion section to strengthen our analysis of study limitations and future research directions.

A more minor point at this stage is that it was not clear if this framework is intended to only apply to one healthcare system (such as Japan, where the authors are based), or more globally. Details of the healthcare system in which this framework could be applied would be helpful for the readers of an international journal.

Response: We have now added this as a limitation for future research in the international context (Line 272-278).

Round 2

Reviewer 2 Report

Thank you again for the opportunity to review this revised paper. The paper has been substantially revised and is now more appropriate for publication consideration. The paper is more appropriately structed as a research report and contains more information that allow me to provide a more detailed review. Upon reviewing the latest version of this paper, I still consider the topic interesting and believe it may be of interest to other readers of this journal. However, I have some comments for the authors to address before recommending for publication. The main issues are some missing information in the methods, and some minor comments elsewhere that aim to add a little extra clarity to their work.  

Specific comments:

Introduction – This is an international journal so need to specify where this is taking place – is this an issue in rural family medicine only in Japan, for instance, or worldwide? The methods indicate the study took place in Japan – so I would consider making the introduction more focused on the Japanese healthcare system, which is quite different from e.g., The United States. Final sentence – I would expand this sentence to better clarify the purpose of the study.  

Methods:

Line 72 – word “of” seems out of place – suggest remove.

Section 2.3 – lines 86-92: This section requires additional details about the semi-structured interviews. I suggest looking at the Consolidated Criteria for reporting qualitative research (COREQ) guidelines, and other published qualitative studies, to help ensure all relevant information is included. For instance, I would expect to see information about how the semi-structured interview script was developed, what questions it contained, were there probing questions, when were the interviews conducted? How did that happen, e.g., was there one researcher who conducted the interviews? Were others present to take notes, etc. were the interviews recorded and transcribed? In what timeframe were the interviews conducted? Etc. would also be useful to know if interviews were conducted in English or Japanese, and any details of translation if necessary.

2.4 analysis – ideally, a second researcher would also have coded the interview transcripts, although I note they did discus and agree themes. Perhaps mention this in limitations.

Results

Line 105: states three themes, but then appears to list out five themes. Looking at figure 2, it seems the 5 themes are organized into two aspects: one contains content, timing, place, and the remaining two themes are separate? It would be helpful if this could be clarified. At the end of the paragraph on line 107, I suggest a sentence to lead the reader into what is coming next. It seems that the authors then talk about each section of the framework next, so a sentence to this effect would be helpful for readers.

Line 122: does it ensure effective learning, or (more appropriately) HELP to ensure effective learning?

Discussion

Seems appropriate

Conclusion:

Would like to see more comment about the framework (since that is the purpose of the study).

Author Response

Responses to the reviewer’s comments

Thank you very much for reviewing our manuscript and providing suggestions for its improvement. We have provided point-by-point responses to the reviewer’s comments; our revisions are indicated in red font in the document. We hope that the revised manuscript meets the journal’s requirements and can now be considered for publication.

Thank you again for the opportunity to review this revised paper. The paper has been substantially revised and is now more appropriate for publication consideration. The paper is more appropriately structed as a research report and contains more information that allow me to provide a more detailed review. Upon reviewing the latest version of this paper, I still consider the topic interesting and believe it may be of interest to other readers of this journal. However, I have some comments for the authors to address before recommending for publication. The main issues are some missing information in the methods, and some minor comments elsewhere that aim to add a little extra clarity to their work.  

 Response: Thank you for the valuable feedback. We have revised the manuscript based on your suggestions.

Specific comments:

Introduction – This is an international journal so need to specify where this is taking place – is this an issue in rural family medicine only in Japan, for instance, or worldwide? The methods indicate the study took place in Japan – so I would consider making the introduction more focused on the Japanese healthcare system, which is quite different from e.g., The United States. Final sentence – I would expand this sentence to better clarify the purpose of the study.  

 Response: Thank you for the valuable feedback. We revised the Introduction by including context regarding Japanese family medicine and clarified the purpose of this study (Line 45 to 57).

Methods:

Line 72 – word “of” seems out of place – suggest remove.

 Response: Thank you for the valuable feedback. We deleted the word.

Section 2.3 – lines 86-92: This section requires additional details about the semi-structured interviews. I suggest looking at the Consolidated Criteria for reporting qualitative research (COREQ) guidelines, and other published qualitative studies, to help ensure all relevant information is included. For instance, I would expect to see information about how the semi-structured interview script was developed, what questions it contained, were there probing questions, when were the interviews conducted? How did that happen, e.g., was there one researcher who conducted the interviews? Were others present to take notes, etc. were the interviews recorded and transcribed? In what timeframe were the interviews conducted? Etc. would also be useful to know if interviews were conducted in English or Japanese, and any details of translation if necessary.

 Response: Thank you for the valuable feedback. We revised section 2.3 comprehensively by including the suggested sections (Line 92 to 104)

2.4 analysis – ideally, a second researcher would also have coded the interview transcripts, although I note they did discus and agree themes. Perhaps mention this in limitations.

  Response: Thank you for the valuable feedback. We revised the limitations by adding the limitation regarding the credibility of this study (Line 287 to 296).

Results

Line 105: states three themes, but then appears to list out five themes. Looking at figure 2, it seems the 5 themes are organized into two aspects: one contains content, timing, place, and the remaining two themes are separate? It would be helpful if this could be clarified. At the end of the paragraph on line 107, I suggest a sentence to lead the reader into what is coming next. It seems that the authors then talk about each section of the framework next, so a sentence to this effect would be helpful for readers.

 Response: Thank you for the valuable feedback. We revised the introduction of the Results by revising “three themes” to “five themes” and adding a sentence leading to the following sections.

Line 122: does it ensure effective learning, or (more appropriately) HELP to ensure effective learning?

  Response: Thank you for the valuable feedback. We revised the phrase following your suggestion.

Discussion

Seems appropriate

Response: Thank you for the valuable feedback.

Conclusion:

Would like to see more comment about the framework (since that is the purpose of the study).

 Response: Thank you for the valuable feedback. We revised the Conclusion, based on your suggestion, by adding an explanation of the framework and its applicability.